# MULTISCALED SOLITARY WAVES

Oleg G. Derzho

Institute of Thermophysics, Russian Academy of Sciences, Novosibirsk, Russia

*Correspondence to:* Dr. Oleg Derzho, oderzho@mun.ca

**Abstract.** It is analytically shown how competing nonlinearities yield multiscaled structures for internal solitary waves in shallow fluids. These solitary waves only exist for large amplitudes beyond the limit of applicability of the KdV equation or its usual extensions. Multiscaling phenomenon exists or does not exist for almost identical density profiles. Trapped core inside the wave prevents appearance of such multiple scales within the core area. The structural stability of waves of large amplitude is briefly discussed. Waves of large amplitude displaying both quadratic, cubic and higher order nonlinear terms have stable and unstable branches. Multiscaled waves without vortex core are shown to be structurally unstable. It is anticipated that multiscaling phenomena exist for solitary waves in various physical origins.

## 1 *Introduction*

The typical horizontal scale (or scales) is a major characteristic of a plane disturbance propagating in a nonuniform medium. Usually, in an ideal density stratified shallow fluid, a wave of small, albeit finite amplitude has one typical scale resulting from the (local) balance between nonlinearity and dispersion like in the realm of Korteveg-de Vries (KdV) equation (Helfrich and Melville , 2006). Solitary waves of permanent form for which capillary dispersion is of the same order as the gravitational one may have oscillatory outskirts as predicted by Benjamin (1992). When viscosity is taken into account, transient effects leading to various length scales are discussed for the KdV type equation with cubic nonlinearity, for example by Grimshaw et al. (2003). In the present note it is shown that for the gravitational dispersion, ignoring all other earlier mentioned effects, solitary waves with multiple scales are possible. These solutions exist only for disturbances of finite amplitude exceeding the range of applicability of the extended KdV model, which incorporates both quadratic and cubic nonlinearities. Higher nonlinearity in the existing small amplitude KdV or mKdV models leads to the correction of the wave length scale without generation of multiscaling. For appearance of multiscaling, the various competitive nonlinearities should be of the same order and that order needs to be higher than the cubic one as analytically discussed below. This effect was initially noticed by Derzho and Borisov (1990) in the Russian journal but the result was not widely disseminated. Recently Dunphy et al. (2011) presented numerical procedure that provides fast calculations for gravitational waves between rigid lids . This model is able to work with fine density stratifications. Dunphy et al. (2011) reported two-humped and usual one-humped solitary internal waves solutions for nearly identical density profiles in a two-pycnocline density stratification. Lamb and Wan (1998) have numerically shown that in some stratifications with two pycnoclines three conjugate flow solutions leading to two-humped solitary waves were present. Makarenko et al. (2009) theoretically considered continuous stratification in order to characterize the role of vertical

structure of the fluid density in the context of waves close to the limiting amplitude. To the best of author knowledge, neither specific nonlinearity in terms of power series of wave amplitude necessary to reveal a two-humped structure nor regions of density profiles with single pycnocline at which such structures exist were not examined in the literature. Kurkina et al. (2011) derived KdV like equation with quadratic and quartic nonlinear terms for interfacial transient waves for the specific three layer geometry. Assumption on small albeit finite wave amplitude was essential to balance nonlinearity and dispersion in that study. Table-top limiting solutions were reported and they were stable within the accuracy of their numerical scheme. In the current paper, an asymptotic model presented in earlier papers by the author addresses multiscaling phenomena for internal solitary waves under free surface in the frame of Dubreil-Jacotin-Long equation (DJL) (Long , 1965). Special attention is given to the case of complicated nonlinearity involving both quadratic, cubic and quartic nonlinear terms for the case of continuous stratification with a single pycnocline . Solitary waves of permanent form, their existence and structural stability are discussed. It is worth noting that family of solutions is richer than two-humped structures. It is expectable that such multiscaled solitary waves exist in other physical systems where complicated competitive nonlinearities are balanced by dispersion.

## 2    *Model for internal waves*

Let us consider the two-dimensional steady motion of an ideal stratified fluid in a frame of reference moving with phase speed of wave $c$ . The approach is asymptotic being based on the DJL equation for waves without a priori limitation on amplitude. This approach has started from the pioneering work by Benney and Ko  (1978). Let us consider the stratification in the form

$$\rho_0\left(z\right) = \rho_{00}(1 - \sigma(z + \delta f(z))), \delta << 1, \sigma << 1, f \sim 1, \tag{1}$$

where $\sigma$   denotes Boussinesq parameter. In (Derzho and Velarde, 1995) it was shown that for this case the dimensionless (primed) streamfunction $\psi' = -\psi/cH$ of a solitary disturbance obeys the equation

$$\psi_{zz} + \mu^2\psi_{xx} + \lambda(\psi - z) - \frac{\sigma}{2}({\psi_z}^2 - 1 - 2\psi\lambda(\psi - z)) + \delta\lambda(\psi - z)f_\psi(\psi) = o(\sigma, \delta, \mu^2), \tag{2}$$

where $\mu$ is the aspect ratio $H/L$ and $\lambda = \frac{\sigma g H}{c^2}$.

In Eq.(2) $z$ denotes the vertical axis, taken positive upwards and $x$ corresponds to the horizontal axis; $z$ and $x$ are scaled with $H$ and $L$, the given vertical and horizontal scales respectively. Expecting no confusion we have, for simplicity, dropped the primes in Eq.(2). Let us locate the bottom and the surface at the dimensionless heights $z = -0.5$ and $z = 0.5 + \eta(x)$, respectively, where $\eta(x)$ denotes surface displacement. The boundary conditions at the bottom and surface are

$$\psi_x = 0 \text{ at } z = -0.5 \tag{3}$$

$$\sigma(\psi_x\psi_z\psi_{zz} - \psi_z^2\psi_{zx}) + \lambda\psi_x = o(\sigma) \text{ at } z = 0.5 + \eta(x) \tag{4}$$

$$\psi_x = -\eta_x\psi_z \tag{5}$$

The solution of Eqs.(2-5) is sought in the form

$$\psi = \psi^{(0)} + \mu^2 \psi^{(1)} + ..., \lambda = \lambda^{(0)} + \mu^2 \lambda^{(1)} + ..., \eta = \eta^{(0)} + \mu^2 \eta^{(1)} + ..., \tag{6}$$

where zeroth order variables are of order unity. Below we shall provide solution for the first mode, which is most frequently observed in nature. The analysis for the higher modes is similar. In the zeroth order

$$\psi^{(0)} = z + A(x)\cos(\pi z), \ \lambda^{(0)} = \pi^2, \eta^{(0)} = 0, \tag{7}$$

where the amplitude function $A(x)$ is to be determined at a higher order. For the solution to the first order equation to exist the solvability condition (Fredholm alternative) demands

$$A_{xxx} + \lambda^{(1)} A_x - \frac{\sigma}{\mu^2}(2A_x - 8\pi A A_x + 2\pi^2 A^2 A_x) + 2\frac{\delta}{\mu^2} Q_x(A) = 0 \tag{8}$$

$$Q(A) = A \int_{-0.5}^{0.5} \cos^2(\pi z) f_\psi(\psi = \psi^{(0)}) dz \tag{9}$$

In order to (locally) balance nonlinearity and dispersion we have to require $max(\sigma/\mu^2, \delta/\mu^2) \sim 1$ thus determining $L$. Benney and Ko (1978) suggested to consider the nonlinear terms as power series in the Boussinesq parameter instead of the small amplitude parameter. Derzho and Velarde (1995) somewhat extended this idea to account a more general undisturbed flow state. After straightforward integrations for remaining solitary wave solution, Eqs.(8-9) can be reduced to

$$A_x^2 + \lambda^{(1)} A^2 + 2\frac{\delta}{\mu^2} \int_0^A Q(A') dA' + A^2 (\frac{8\pi A}{3} - 2 - \frac{\pi^2 A^2}{3}) = 0. \tag{10}$$

The Weierstrass approximation theorem states that every continuous function defined on a closed interval can be uniformly approximated as closely as desired by a polynomial function. Recent account on the topic is reviewed in Hazewinkel (2001). Thus, the integral below can be represented with the help of some $N$th-order polynomial according to the Weierstrass approximation theorem. In the current study only polynomial formula for stratification is considered, thus it directly leads to nonlinearities in the polynomial form.

$$\int_0^A Q(A') dA' = A^2 P_N(A), \tag{11}$$

For the wave of amplitude $A_0$ Eq.(10) yields

$$\frac{A_x^2}{A^2} = (A_0 - A)\Phi(A, A_0), \tag{12}$$

$$\tag{13}$$

$$\Phi(A, A_0) = 2\frac{\delta}{\mu^2}\frac{P_N(A_0) - P_N(A)}{A_0 - A} + \frac{\sigma}{\mu^2}\frac{\pi^2}{3}(\frac{8}{\pi} - A - A_0) \tag{14}$$

$$\lambda^{(1)} = \frac{\sigma}{\mu^2}(-\frac{8\pi A}{3} + 2 + \frac{\pi^2 A^2}{3}) - 2\frac{\delta}{\mu^2}P_N(A_0) \tag{15}$$

Equations (12-14) determine completely both profile and phase velocity of a solitary wave with amplitude $A_0$.

## 3  *Multiscaling*

The function $f$ in the form of a $M$th-order polynomial generates $P_N$ with the index $N = M - 1$. The power index of $\Phi$ is thus $max(1, M - 2)$. The condition for Eq.(8) to possess a multiscaled solution reduces to the condition that $\Phi(A, A_0)$ must
be sign-defined with several extrema within $[0, A_0]$ . Thus it must have more than two imaginary roots on that interval. It determines $M \geq 4$ , i.e. for a stratification in the form of cubic polynomial or if wave amplitude is small enough to neglect $A^4$ and higher order nonlinearities, multiscaled solitary waves do not exist because $f$ has no imaginary roots for this case. This is why classical KdV or mKdV can not provide multiscaled solitary waves over flat bottom. Let us consider wave structures for the density stratification in the form,

$$\rho_0(z) = \rho(1 - \sigma z + 0.5\sigma^2 z^2 + \alpha\sigma^2 z^4), \tag{16}$$

which produces quadratic, cubic and quartic terms in Eq.(8). Thus Eq. (12) for this case of stratification becomes

$$\Phi(A, A_0) = \frac{\sigma}{\mu^2}\left[\frac{-8\pi}{3}(\frac{1}{3} + 2\alpha - \frac{160\alpha}{9\pi^2}) + \frac{\pi^2}{3}(A + A_0) + \frac{128\alpha\pi^2}{75}(A^2 + A_0^2 + AA_0)\right] \tag{17}$$

Two humped solitary waves for the stratification given by Eq.(17) exist in the domain shown in Fig.1.

Two-humped solitary wave with amplitude $A_0$= 0.1885 for the particular stratification profile Eq.(16) with $\alpha = -1.39$ and
$\sigma = 0.01$ is shown in Fig. 2 and Fig. 3.

Indeed, the maximum derivative on $x$ in the dimensionless coordinates is of order unity. However, the wave has a pronounced two-scale structure with typical length scales, which are much larger than the length $L$ used to scale the derivative. A solitary wave with three typical length scales ( three-humped one) is shown in Fig. 4.

For this case the stratification profile is

$$\rho_0(z) = \rho(1 - \sigma z + \sigma^2(1.206z^2 - 4.37z^3 - 3.435z^4 - 33.407z^6)), \tag{18}$$

which produces in Eq. (8) nonlinear terms up to $A^6$. Generally, one can expect at most $M/2$ different scales for a stratification in the form of polynomial with even power index $M$, and $(M - 1)/2$ otherwise.

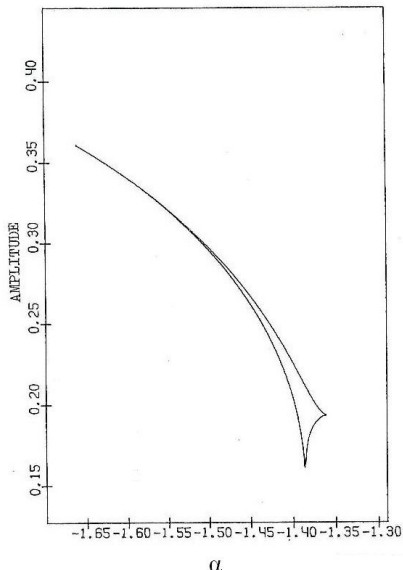

**Figure 1.** Existence domain for two-humped solitary wave.

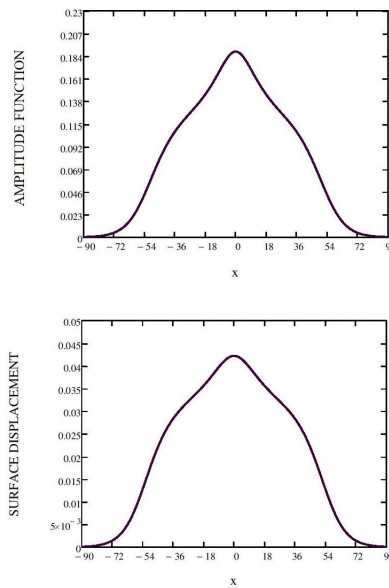

**Figure 2.** Amplitude function and surface displacement for two-humped solitary wave, $\alpha = -1.39$.

Further, we wish to examine the structure of solitary waves of permanent form for the stratification given by Eq.(16). We only consider the case $\alpha = -1.39$ and focus on the waves of permanent form under free surface, their domain of existence,

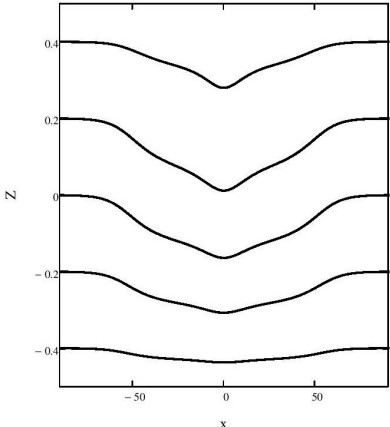

**Figure 3.** Streamlines for two-humped solitary wave, $\alpha = -1.39, A_0 = 0.1885$.

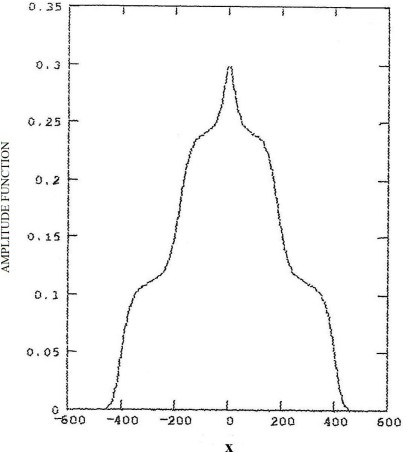

**Figure 4.** Amplitude function for three-humped solitary wave.

limiting forms and structural stability. Other values of $\alpha$ lead to more extensive consideration with a number of particular cases. Such study is beyond the scope of the present paper. First, for $\alpha = -1.39$ there exist only permanent waves with positive amplitudes. Wave phase velocity is defined by the following expression

$$5 \quad c^{(1)}(A_0) = \frac{c - c(A_0 = 0)}{\mu^2} = \frac{4A_0}{3\pi}(2\alpha + \frac{1}{3} - \frac{160\alpha}{9\pi^2}) - \frac{A_0^2}{6} + \frac{64\alpha A_0^3}{75\pi}, \tag{18}$$

Fig.5 shows that the phase velocity is an increasing function for $0 < A_0 < A_2$ and $A_0 > A_1$. For $A_1 < A_0 < A_2$ the phase velocity decreases with amplitude and there are no steady solitary wave solutions. When $0 < A_0 < A_2$ solitary waves are

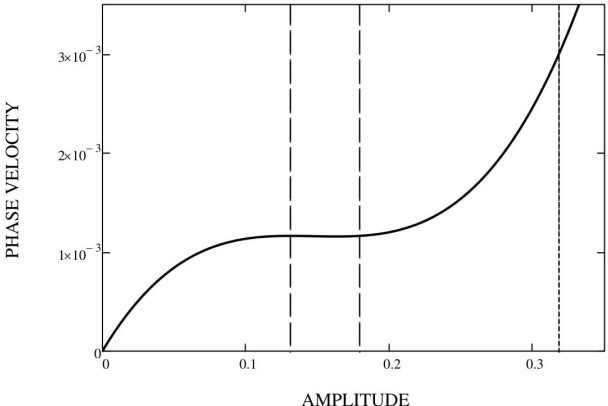

**Figure 5.** Phase velocity versus wave amplitude. Solid line - $c^{(1)}(A_0$ , dashed lines $0 < A_0 = A_1$ and $A_0 = A_2$, dotted line $A_0 = A_* = 1/\pi$, critical amplitude above which the model does not work as a vortex core arises inside the wave. $A_2 = 0.1311, A_1 = 0.1793$.

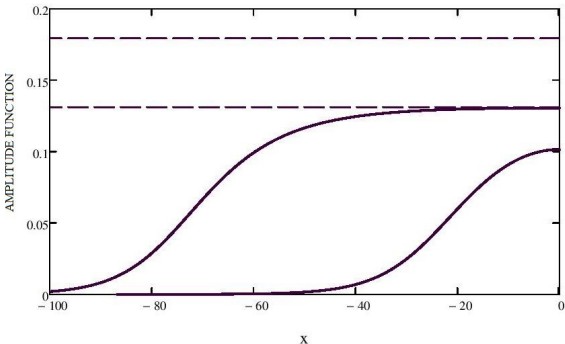

**Figure 6.** Profiles of stable solitary wave are shown by solid lines. Dashed lines correspond to $A_2 = 0.1311, A_1 = 0.1793$. Limiting amplitude reaches when $A_0 = A_2$.

widening as amplitude increases with a table top limiting shape with a local maximum for the wave velocity as shown in Fig.5 and Fig.6.

Such waves are structurally stable according to Bona et al. (1987) as both the wave energy $E = \int_{-\infty}^{\infty} A^2 dx$ and the wave velocity increase as amplitude increases.

For $A_0 > A_1$ wave profiles are shown in Fig.7. Waves change from the table top solution to solitary waves with a single scale via multiscaled structures.

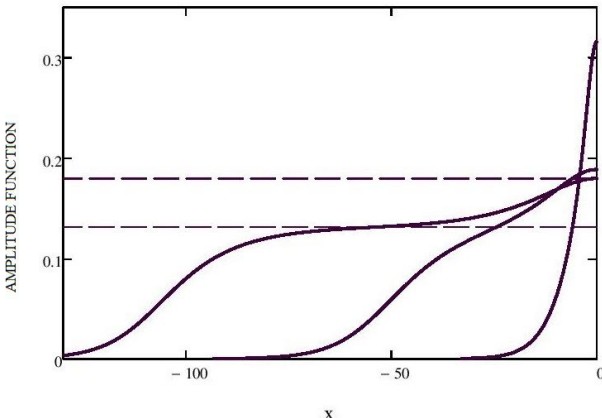

**Figure 7.** Profiles of unstable solitary wave are shown by solid lines. Dashed lines $A_2 = 0.1311$, $A_1 = 0.1793$. Lower limiting wave amplitude is $A_0 = A_1$.

For the particular stratification considered here waves are structurally unstable (Bona et al., 1987) since wave energy decreases as shown in Fig.8 but the wave velocity increases with the increasing of wave amplitude. Interesting observation is these waves of sufficiently large amplitude could be stable as the energy is eventually increased as shown in Fig.8. For the stratification considered here, it does not matter because solution with vortex core appears at lower amplitude when energy is still the decreasing function of amplitude. However it leads to the interesting phenomenon - waves with vortex core could stabilise the wave. The idea is that the vortex core leads to widening of wave (Derzho and Grimshaw , 1997) and consequently to the increase of its energy, thus the structural stability criterion will be satisfied. For the considered particular stratification waves with vortex core are initially unstable as increase of energy due to the vortex core and associated widening does not compensate the decrease of energy in the wave outside the vortex core. Nonetheless, above some amplitude waves become structurally stable. When wave amplitude further increases the permanent wave of limiting amplitude becomes infinitely wide as shown by Derzho and Grimshaw (1997).

The theory described above is valid for wave amplitudes below $A_*$, a certain amplitude at which a vortex core started to appear inside the wave. For nearly linear density profile $A_* = 1/\pi$. Derzho and Grimshaw (1997) have shown that

$$B_x^2 \sim R(A_*)(1-B) - \frac{8\nu}{15}(1-B^{5/2}), \tag{19}$$

where $\nu$ is the supercriticality parameter defined such that $B$ varies from zero to one as wave amplitude does from $A_*$ to the maximum value allowed predicted there. $R(A_*)$ depends on stratification profile and is fixed. It is straightforward to notice that $B(x)$ is monotonic and therefore multiscaling in the vortex core area does not exist when $A > A_*$.

Multiscaling effects similar to the discussed above, could be observed in various physical media. Derzho and Grimshaw (2005) reported that solitary Rossby waves in channels obey the same KdV type equation with complicated nonlinearity due

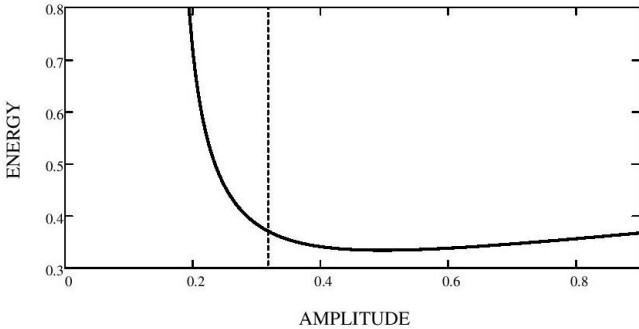

**Figure 8.** Energy versus wave amplitude. Dotted line corresponds to $A_0 = A_*$.

to the mean shear variations. Coriolis force for Rossby waves plays the same role as gravitational force for the internal gravity waves. The results on multiscaling for Rossby waves with and without trapped core will be reported elsewhere.

**4   *Conclusion***

For the particular case of a nonlinear dispersive medium like a shallow stratified fluid embedded in the gravity field, we have addressed multiscaled solitary waves which are predicted when there exists competition of several different types of nonlinearity. The mechanism leading to these solutions differs from the mechanism of multiscaling due to the competition of different types of dispersion or effects due to the dissipation. We have shown that the length used to scale the $x$-derivative does

not simply coincide with the typical length scale of the wave, as for KdV. Moreover, multiscaled (multi humped) disturbances exist for sufficiently large amplitudes, at least terms in fourth order of waves amplitude should be accounted. Multiscaling (multi humped) phenomenon exists or does not exist for almost identical density profiles, two pycnoclines case studied earlier is not necessary for the existence of multiscaling. Continuous stratification given by Eq.(16) was studied in more detail. The structure of permanent solitary waves and how multiscaling appeared were presented. Structural stability was examined using

the criterion proposed by Bona et al. (1987). It was shown that both stable and unstable solutions of the KdV type equation with quadratic, cubic and quartic nonlinearities are available. Multiscaled waves without trapped core belong to the unstable solutions. Trapped core inside the wave prevents appearance of such multiple scales within the core area. However, trapped core could stabilise the multiscaled solution in the sense of structural stability. The case when trapped core and multiscaling are combined together is beyond the scope of the present study and will be presented elsewhere. It is noted that multiscaling phenomena exist for solitary waves in various physical origins, for example, for Rossby waves on a shear flow (Derzho and Grimshaw, 2005) or inertial waves in swirling flows (Derzho and Grimshaw, 2002).

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
