# Peer review of "MULTISCALED SOLITARY WAVES"

_Nonlinear Processes in Geophysics, 2017_

## Referee Comment (RC1) · M. Stastna (Referee) · 8 Mar 2017

This manuscript considers the topic of multi-scaled solitary waves in continuously stratified fluid. The topic is of mathematical interest, though no evidence that such waves actually occurring in nature is provided (or exists in the literature, to the best of my knowledge). The presentation varies between full disclosure of equations, to wide gaps in logic and some very odd, and at times hilarious, linguistic mis-steps. The expression "tree-humped solitary waves" that slipped through the spell checker will linger in my memory for some time.

I have worked on fully nonlinear solitary waves for a very long time. As near as I can understand the context of the results presented (and the presentation of context is pretty poor in this manuscript) it is that for a linear stratification the well known Dubreil-Jacotin-Long (DJL) equation that governs fully nonlinear solitary waves linearizes and no solitary wave solutions are possible. This does not mean that the stratified Euler

equations to which the DJL is equivalent linearize in this case, but it does mean the nonlinearity needs to be addressed by other means (see Grimshaw and Zengxin, JFM 229, for an example which shows why the KdV is not the relevant equation here). The present manuscript considers stratifications that are nearly linear, and uses a perturbation expansion to construct solutions. The author also does not provide any of the context I have provided in the above paragraph and indeed presents the DJL equation as his own past work. I am 100% OK with the author disagreeing with me, but I am not OK with nothing being said at all.

The stratification used for the primary example in Figure 2 has a largest departure from the linear density profile on the order of 5e-6 (or 5e-4 when scaled by the top to bottom density difference). This strikes me as linear for all intents and purposes, and certainly to the extent that field measurements could discern. The author makes no effort to explain how broad of a range of stratifications his theory applies to, and the reference list is 40% self-citation, which would be fine for a strong result, but seems like a poor choice for what looks like a mathematical oddity at best. When I put the stratification used to produce Figure 2 into my DJL solver I do not get multi-scale solitary waves, but a small solitary wave of depression. Lamb and Wan have considered stratifications with multiple solitary waves possible for a given stratification, so I am not discounting the possibility of the multi-scale solitary wave, but it is troubling that the variational method more naturally converges to a different wave.

Incidentally the DJL code I mention above is the same code that we reported in Dunphy et al 2011, and the main point there was not that multi-scale solitary waves actually occur in nature, but that the spectral methods we implemented allow for even something as finely balanced as one of these waves to be computed in minutes. Indeed, as appears to have been missed by the present author, the method outlined in that paper computes exact multi-scale solitary waves for a much broader set of stratifications than the present manuscript addresses. I do want to note that I like the characterization in terms of the polynomial the author provides, but the presentation needs to make the

method reproducible by the reader (at present I have no idea how P_N is computed and the 1968 Mathematical Handbook the author quotes for the result is not useful for providing this vital information).

In order for the manuscript to be publishable the author needs to provide a fair assessment of the literature and his findings. I think some of my comments above, and the detailed comments below, will provide the means to do this. In the end, I think NPG is a good place for nifty mathematical problems but the presentation and editorial standard needs to be much higher than that in the initial version of this manuscript.

Detailed comments:

page 1

Line 10: I think this sentence is meant to say the opposite of what it actually says

Line 12: How can a similar effect be observed due to viscosity. Viscosity means energy is not conserved and hence solitary waves cannot exist.

Line 23: again I think the sentence states the opposite of what it actually means to state

page 2:

The equation (number 2) is the DJL equation, why not explicitly state this?

What is the reason for keeping the free surface? It seems like an unnecessary complication.

Line 20: "searched" is not the correct verb here; perhaps "sought"?

Equation (7) and similar expressions; please use \cos in latex

page 3:

Line 20: So the whole set up is a perturbation of the linearly stratified case? Seems restrictive. Then the solvability condition is expressed in terms of a polynomial P_N

which is only given implicitly? An example or two here is essential.

page 4: "tree-humped"

Figure 1 is hard to make out, but I guess the ordinal is alpha (written as "alfa"), the definition of which only appears after Figure 1 is discussed. Or is this the delta of equation (1)? In any event, a clearer exposition is needed.

page 5 and 6: The Conclusions are really barebones. Is it possible to suggest how these waves could be generated; would flow over topography do it?

---

## Author Comment (AC1) · 25 Mar 2017

**Dear Prof. Stastna,**

**Thanks for your comments**. **My point to point replies are written in bold for better readability.**

**General remarks.**

1.  The presentation varies between full disclosure of equations, to wide gaps in logic and some very odd, and at times hilarious, linguistic mis-steps.

    **Words such as hilarious and odd, wide logic gaps are not appropriate according to the Editorial Policy.**

2.  As near as I can understand the context of the results presented (and the presentation of context is pretty poor in this manuscript) it is that for a linear stratification the well known Dubreil-Jacotin-Long (DJL) equation that governs fully nonlinear solitary waves linearizes and no solitary wave solutions are possible. This does not mean that the stratified Euler equations to which the DJL is equivalent linearize in this case, but it does mean the nonlinearity needs to be addressed by other means (see Grimshaw and Zengxin, JFM 229), for an example which shows why the KdV is not the relevant equation here.

    **It is correct that DJL equation for exactly linear stratification is linear even if wave amplitude is not small. Grimshaw and Zengxin (JFM, 1991) derived the forced Korteweg-de Vries equation to describe resonant flow over topography. When the fluid is uniformly and weakly stratified the quadratic nonlinear term is absent. That case requires an alternative theory which was examined in the above mentioned paper. It is crucial to note that the case considered in the submitted manuscript implies that**
    **a) the rigid lower boundary is flat, so flow over topography is irrelevant to the present manuscript;**
    **b) stratification is essentially nonlinear as stated on page 2, eq.(1). So nonlinearity is present and this nonlinearity is responsible for the multiscaling. It is the core point of the paper.**

3.  The author also does not provide any of the context. I have provided in the above paragraph and indeed presents the DJL equation as his own past work.

    **I certainly do not present DJL equation as own past work. The paper clearly states that on page 3 lines 7 and 8. I will add the appropriate references on pages 1 and 2 to avoid any misunderstanding. Additionally I would add more references on the subject of the paper as suggested.**

4.  The stratification used for the primary example in Figure 2 has a largest departure from the linear density profile on the order of 5e-6 (or 5e-4 when scaled by the top to bottom density difference). This strikes me as linear for all intents and

purposes, and certainly to the extent that field measurements could discern. The author makes no effort to explain how broad of a range of stratifications his theory applies to.

**The presented theory is asymptotic so the range of stratifications the theory applies to is given in the model assumptions. Please look at Eq. (1). Stratification is essentially nonlinear as stated on page 2, Eq.(1). Effects of nonlinearity associated with nonlinear stratification lead the multiscaling. It is the main point as anticipated above.**

**Variation of stratification due to the fine structure mentioned in Eq. (15) is $2\alpha\sigma^2 \sim 5*10^{-4}$ as noticed by the referee. From this figure the referee stroked the profile "as linear for all intents and purposes, and certainly to the extent that field measurements could discern". To this point, I would again mention that the paper considers asymptotic produre and some numbers are immaterial and presented for illustrational purposes. The major point of any asymptotic theory is the definition of scales. The value of $\sigma$ defines wave speed according to the equation in line 10, page 2. The value of $\delta$ defines the horizontal length scale once we fix $\delta/\mu^2$. For illustration $\sigma = 0.01$ and $\delta = \sigma$ were taken in the manuscript. For example, if $\sigma = 0.04$ and $\delta = 0.25$ the result of the manuscript remains the same, just wave speed is doubled and physical horizontal length scale is increased in 5 times.**

5. When I put the stratification used to produce Figure 2 into my DJL solver I do not get multi-scale solitary waves, but a small solitary wave of depression. I am not discounting the possibility of the multi-scale solitary wave, but it is troubling that the variational method more naturally converges to a different wave.

   **As I understand your DJL solver, it is not designed to account for free surface and the nonlinearity associated with it. Thus the comparison with the presented model is not correct. Moreover, I just have to notice that any numerical matters related to the specific numerical method with its specific convergence are beyond the scope of the present manuscript. However, I definitely mention the spectral method implemented in your group that is designed to be applicable to a broader set of stratifications even with fine structure.**

6. I do want to note that I like the characterization in terms of the polynomial the author provides, but the presentation needs to make the method reproducible by the reader (at present I have no idea how P_N is computed and the 1968 Mathematical Handbook the author quotes for the result is not useful for providing this vital information).

   **In the current manuscript only polynomial formula for stratification is considered, it directly leads to nonlinearities in the polynomial form. To justify the model for a more complicated form of stratification the**

**Weierstrass approximation theorem (1885) provides a theoretical foundation for the presented approach. The Weierstrass approximation theorem states that every continuous function defined on a closed interval [a,b] can be uniformly approximated as closely as desired by a polynomial function. ( for recent accounts on the topic look at Hazewinkel, Michiel, ed. (2001), "Stone-Weierstrass theorem", Encyclopedia of Mathematics, Springer, ISBN 978-1-55608-010-4). I did not construct approximations in the paper, just mentioned that it is a mathematically correct procedure. Such construction is beyond the scope of the present manuscript.**

7. The reference list is 40% self-citation, which would be fine for a strong result, but seems like a poor choice for what looks like a mathematical oddity at best.

   **The list of references will be extended in the revised version. Nonetheless, this "oddity" was discussed in your own paper. The major point is that the present asymptotic model predicted multi humped solitary waves back in 1990 and the original result was cited only once in 2011 in NPG, nothing appeared before and after. In order to demonstrate priority, the author decided to submit to NPG as to an open access journal for wide international audience.**

**Specific detailed comments.**

page 1

"Lines 10 sentence is meant to say the opposite of what it actually says"

**Line 10 probably reads unclear. A capillary ripple superimposed on gravity wave is one of such examples. Going to change as follows.**
**If the wavelength of the disturbance is too small AND COULD DISPLAY, for instance, capillary dispersion, multiscaled solitary waves are possible as shown by Benjamin (1992).**

Line 12: How can a similar effect be observed due to viscosity. Viscosity means energy is not conserved and hence solitary waves cannot exist.

**I did not mean stationary solitary waves in that sentence. I meant the situation in the vicinity of the breaking point where singularity resolves through the generation of an oscillatory zone, weak dissipation defines the scales in that zone as originally described by Benjamin & Lighthill 1954. Multiscaling in Introduction means that several scales could be observed due to different competing physical mechanisms contrary to the one physical nonlinearity described later on in the paper**.

Line 23: again I think the sentence states the opposite of what it actually means to state.

**In the sentence appeared in lines 23-25 the word "Neither" should be replaced with "Either"   The additional letter "N" was a sad typo. Sorry about that.**

**The sentence now reads as follows.**
**However, either specific nonlinearity in terms of power series in wave amplitude necessary to reveal a two humped structure or regions of density profiles at which such structures exist were not presented.**

page 2:
The equation (number 2) is the DJL equation, why not explicitly state this?

**I will state this for sure. However eq.(2) is written readily for asymptotic theoretical modelling using the proposed assumptions. For DJL in its full form an application of asymptotic approach is impossible.**

What is the reason for keeping the free surface? It seems like an unnecessary complication.

**The reason why the free surface is kept is that I presented the effect of the multiscaling on the surface, i.e. predicted scale and height of the surface displacement. Free surface also affects nonlinearity in the solitary wave. To this end, direct comparison with DJL numerical models between solid boundaries is incorrect as mentioned before.**

Line 20: "searched" is not the correct verb here; perhaps "sought"?  **Agree, will change**.

Line 20. Equation (7) and similar expressions; please use \cos in latex.
**Agree, will change.**

page 3:

Line 20: So the whole set up is a perturbation of the linearly stratified case? Seems restrictive.

**Do you mean the perturbation as a flow that returns to a linearly stratified state when the wave has passed through? It is not the case since undisturbed stratification is nonlinear as stated in eq. (1). However, undisturbed stratification is supposed to be only slightly nonlinear. Yes, it is a restriction. But this restriction introduces a small parameter necessary to construct a tractable asymptotic model without restriction of small wave amplitude. And the presented approach generalises the seminal work by Benney and Ko (1978)**

Then the solvability condition is expressed in terms of a polynomial P_N

which is only given implicitly? An example or two here is essential.

**In the revised paper I will present the amplitude equation in exact form for the specific stratification given in eq. (15)**

page 4: "tree-humped"   **Sorry for the typo that slipped through the spell checker.**

Figure 1 is hard to make out, but I guess the ordinal is alpha (written as "alfa"), the definition of which only appears after Figure 1 is discussed. Or is this the delta of equation (1)? In any event, a clearer exposition is needed.

**Alpha is a constant as defined in eq. (15) I need to mention that below the eq.(15. The value of $\delta$ defines the horizontal length scale once we fix $\delta/\mu^2$. For illustration $\sigma = 0.01$ and $\delta = \sigma$ were taken in the manuscript. Moreover, Lamb and Wan (1998) and Damphy et al. (2011) numerically discussed the case of 2 pycnoclines stratification. The present study (along with the earlier paper by the author) explicitly shows multiscaling for  the case of monotonic Brunt-Vaisala frequency, see eq. (15).**

**The figure contains a typo and will be corrected in the revision. Sorry about that.**

page 5 and 6: The Conclusions are really barebones. Is it possible to suggest how these waves could be generated; would flow over topography do it?

**Generation of waves by topography is an essentially transient phenomenon and to this end lies beyond the validity and the scope of the present study. Seminal paper by Benney and Ko (1978) among others stated that any initial disturbance will be split into solitary waves and continuous spectrum. Solitary waves propagate until viscosity effects become apparent.  As derived, for example, by Grimshaw and Yi (1991) uneven topography does not produce nonlinear terms in the forced KdV type equation. The present paper shows that multiscaling is the interplay of various nonlinear terms in the KdV type equation**.

---

## Referee Comment (RC2) · Anonymous Referee #2 · 28 Mar 2017

Author studies the structure of the solitary wave in the weakly stratified fluid on density. If the difference in the density profile from the linear one can be presented by the high-order polynomial. If all terms of this polynomial have the same order, the solution of the Dubreil-Jacotin-Long equation can describe the multiscalled solitary waves. Author also discusses how this result can be obtained from the high-order Korteweg-de Vries equation but KdV-like equation is not derived. This result is interesting, meanwhile, I have some comments.

First of all, author says that usually the KdV-like equation contains quadratic and cubic nonlinearities, and highest nonlinear terms can be neglected. In fact, the analysis of the three-layer fluid demonstrates that such both nonlinearities can be small and here highest nonlinearity should be accounted. In particular, the specific "2+4" KdV-like equation has been derived with use of asymptotic procedure for waves in three-layer fluid in papers:

[Figure]

Kurkina O.E., Kurkin A.A., Soomere, T. Pelinovsky E.N., Ruvinskaya E.A. Higher-order (2+4) Korteweg-de Vries - like equation for interfacial waves in a symmetric three-layer fluid. Physics Fluids. 2011, vol. 23, 116602.

Kurkina O. E., Kurkin A.A., Ruvinshaya E.A., Pelinovsky E.N., Soomere T. Dynamics of solitons in a nonintegrable version of the modified Korteweg – de Vries equation. JETP Letters, 2012, vol. 95, No. 2, 91-95.

In general, it contains all nonlinear terms of the same order, but authors of cited paper analyzed in details a case when only cubic and quantic nonlinearities are keeping in the KdV-like equation. The structure of the solitary waves in the KdV-like equation with all nonlinear terms up to 5th order is also discussed in two papers:

Poloukhina O.E., Pelinovsky E.N., Slunyaev A.V. Extended Gardner equation for internal waves in stratified fluid. Institute of Applied Physics, Preprint, 2002.

Poloukhina O.E., Slunyaev A.V. Extended evolution model based on the Gardner equation for internal waves in stratified fluid. Izvestiya, Academy of Engineering Science, 2006, vol. 18, 82 – 90.

(Both last papers are in Russian and I ask Efim Pelinovsky to send these papers directly to author). The solitary waves interact inelastically. All such models are not integrable and they lead to instability of waves of large amplitudes.

Instability of solutions of the "n-nonlinear" KdV equation (with term $u^n$) for $n > 4$ is well-known, and it is evident for me that such instability should be in Derzho's model which is not discussed. This should be clarified in revised paper.

Minor remarks:

Page 1: line 12 - may be useful to add the reference as Grimshaw R., Pelinovsky E., Talipova T. Damping of large-amplitude solitary waves. Wave Motion, 2003, V. 3, No. 4, 351 - 364 where the various kinds of damping terms and their influence on the shapes of the solitary wave are studied.

page 4: Fig.1 - the title of axes "alfa" is plotted instead of "alpha" Fig.2 - to add the title in the vertical axes

line 10 - may be "three - humped"?

page 5: Fig.3 - to add the titles in both axes

I recommend to accept the paper after the revision.

---

## Author Comment (AC2) · 2 May 2017

Dear Referee 2, Thanks for the fruitful comments. All minor comments and appropriate references to solitary waves with 2+4 non-linearity are accepted and will be incorporated in the revised version. Major comment on stability will be discussed. Actually the revised version will include stability analysis for the considered waves, main result is that some 2 +3+4 waves are stable but some do unstable. The revision with detailed maps for existence of permanent solutions and their stability will be presented. The revision will be submitted shortly. Dr. O. Derzho

---

## Author Response (AR2)

**I would like to thank the referees for the fruitful comments.**

**Referee 2**

It seems the title for the vertical axis in Figs 2 (up) and 4 as "amplitude function" should be better, because of "amplitude" is the maximal point on the wave shape, and you use this term "amplitude" for A0 in text.

**Thanks, agree, changed to "Amplitude function" where needed (Figs 2upper,4,6, and 7).**

**Referee 1**

I worked quite hard to try to make this happen with the parameters the author provides (and is to be commended for taking care to ensure this is the case), and was not successful. It is possible that the free surface which the author accounts for in his asymptotic procedure, but which is absent in the formulation of the DJL equation I use, is essential for this, but I cannot see a physical reason why.

**There is the physical reason which is discussed in any classical course on internal waves. The reason is that the boundary conditions at the free surface add additional nonlinearity. That is why the numerical models for waves propagating between rigid lids on which you are working do not repeat my results.**

The manuscript, in my opinion, starts to drift when the topic of trapped cores is brought up. Trapped cores either invalidate the assumptions used to derive the DJL equation, or are a dynamical feature that naturally evolves, for example when a wave shoals. They have been extensively discussed in the literature, and none of this discussion appears in either the text or the references. I understand the author wants to present his ideas about wave stability and hence is not obliged to fully review the literature, but some commentary would help the reader orient the present study.

**I believe that any additional discussion about transient effects and shoaling is beyond the scope of the present study. The study on the waves with vortex core has been cited in the paper.**

I thus hope the author will adopt a subset of the suggestions below, and after this I think the manuscript can be published.

**Thank you; please find my answers to the minor suggestions below.**

1) The title seems excessively general. At the very least "in a stratified fluid" should be added.

**The article refers not only to stratified flows. Similar result is valid for Rossby waves and inertial waves in swirling flows so the title reflects this point. Appropriate reference has been added.**

2) "then" refers to a comparison in time as in "I ate lunch and then I ate supper". "than" is the correct word when the author states "the family of solutions is richer than two-humped structures". A similar error occurs at other points in the article.

**Sorry for that typo. Corrected.**

3) The description of what we did in Dunphy et al isn't quite right. It might make sense to describe Lamb and Wan's work first since their result is what allowed us to construct the multi-scaled solitary wave solution. The part about nearly identical profiles isn't really relevant to that aspect of the study. I suppose the author feels it is important to mention since he argues that very small differences in density profiles can make the difference in whether multiscaled waves do or do not exist in his formulation. In the two pycnocline example from Lamb and Wan we were following, small differences in stratification made no difference to the calculation.

**I implemented the suggestion given in your initial comment given below**
**"I mention above is the same code that we reported in Dunphy et al 2011, and the main point there was not that multi-scale solitary waves actually occur in nature, but that the spectral methods we implemented allow for even something as finely balanced as one of these waves to be computed in minutes"**

**The major point is that Dunphy et al 2011 presented a numerical code able to deal with fine details of stratification. That is why I referred this paper first. I dealt with situation without 2 pycnoclines but with the fine structure of stratification as discussed in the revised paper. I am hoping it explains my presentation.**

4) On page 3 the discussion of the Weierstrass approximation theorem has been expanded but I still think it's a bit unclear. At the very least the author should state that f(z) is now taken to be a polynomial (may be before equation (11)). It would be helpful to tell the reader whether going from the general expression (11) to the specific result (17) is algorithmic, or whether it just kind of worked out for this choice of f(z).

**It is clearly explained in the paper that I presented all formulae for the wave characteristics based on the polynomial profile of stratification.**

I also think the section heading "Multiscaling" occurs in an odd place, since the first sentence ties in very nicely to the last pargraph of the previous section.

**The last paragraph before the "Multisaling" section presents the general result. All material presented in "Multiscaling" related to the specific cases when multiscaling**

**was found. The word "pargraph" in the referee review contains a typo. We are all making such mistakes.**

Finally, it would be very useful to have a table with f(z), P_N(z) for the various special cases discussed, possible with a column for relevant figures.

**Tables are necessary if no formula can be presented. I presented all assumptions and formulae necessary to calculate the wave characteristics**.

5) Is phase velocity the correct term? I am not aware of a group velocity for solitary waves, so wouldn't propagation speed be easier to understand?

**Phase velocity is a classical definition discussed in any course on nonlinear waves so the term should be used as is.**

6)  Both the figure captions, and the discussion of the figures in the text are very brief. Figure 1 is very useful. I would tighten the axes to show how special the region required really is, and I would add a vertical line at the value of alpha used on page 4. Then I would describe this in the text.

**I believe that a brief discussion is in line with the length of this short paper. The referee agreed that the length of the paper is appropriate and ticked the relevant line in the assessment of the paper.**

7) On page 7 the author states that waves "evolve". This is misleading, since there is no temporal evolution, merely a tracing of the form that the wave takes in parameter space.

**The word "evolve" is replaced by  "change".**

English: "In a Russian journal", "dissimilated" should be changed to "disseminated" or some similar word, "Assumption of small, albeit finite", "a priori" not "a priory", "Multisclaed" not "Multscaled" , "fourth order" not "forth order"

**Corrected. The word "Multisclaed" in the referee review contains a typo. As I mentioned before such small mistakes are done by everybody.   The expression "small, albeit finite amplitude" was used, for example, in the book "IUTAM Symposium on Laminar-Turbulent Transition and Finite Amplitude Solutions edited by Tom Mullin, R. R. Kerswell."**

**Finally, the list of corrected typos is**

1) **"than" line 20, p.1 and line 11 p.2.**
2) **"change" line 6, p.7.**
3) **"disseminated" line 21, p.1.**
4) **"a priori" line 15, p.2.**
5) **"fourth order" line 11,p.9.**